# Nudge strategies for behavior-based prevention and control of neglected tropical diseases: A scoping review and ethical assessment

**Fiona Vande Velde** [1]*, **Hans J. Overgaard** [2,3], **Sheri Bastien** [1,4,5]

1 Department of Public Health Science, Faculty of Landscape and Society, Norwegian University of Life Sciences, Ås, Norway, 2 Faculty of Science and Technology, Norwegian University of Life Sciences, Ås, Norway, 3 Department of Microbiology, Faculty of Medicine, Khon Kaen University, Khon Kaen, Thailand, 4 Department of Community Health Sciences, Cumming School of Medicine, University of Calgary, Calgary, Canada, 5 The Centre for Evidence-Based Public Health: A JBI Affiliated Group, Department of Public Health Science, NMBU, Ås, Norway

* Fiona.vande.velde@nmbu.no

**Data Availability Statement:** All relevant data are within the manuscript and its Supporting Information files.

## Abstract

### Background

Nudging, a strategy that uses subtle stimuli to direct people's behavior, has recently been included as an effective and low-cost behavior change strategy in low- and middle- income countries (LMIC), targeting behavior-based prevention and control of neglected tropical diseases (NTDs). The present scoping review aims to provide a timely overview of how nudge interventions have been applied within this field. In addition, the review proposes a framework for the ethical consideration of nudges for NTD prevention and control, or more broadly global health promotion.

### Methods

A comprehensive search was performed in several databases: MEDLINE, PsycINFO, and Embase (Ovid), Web of Science Core Collection, CINAHL, ERIC and Econ.Lit (EBSCO), as well as registered trials and reviews in CENTRAL and PROSPERO to identify ongoing or unpublished studies. Additionally, studies were included through a handpicked search on websites of governmental nudge units and global health or development organizations.

### Results

This scoping review identified 33 relevant studies, with only two studies targeting NTDs in particular, resulting in a total of 67 nudge strategies. Most nudges targeted handwashing behavior and were focused on general health practices rather than targeting a specific disease. The most common nudge strategies were those targeting decision assistance, such as facilitating commitment and reminder actions. The majority of nudges were of moderate to high ethical standards, with the highest standards being those that had the most

**Funding:** This review was conducted as a part of the MY-SCHOOL project (GLOBVAC Project no. 281588). All authors, FVV, HJO, SB, participated in the project and received funding from the Research Council of Norway, www.forskningsradet.no/en/, under GLOBVAC Project no. 281588. The funders had no role in study design, data collection and analysis, decision to publish, or preparation of the manuscript.

**Competing interests:** The authors have declared that no competing interests exist.

immediate and significant health benefits, and those implemented by agents in a trust relationship with the target audience.

## Conclusion

Three key recommendations should inform research investigating nudge strategies in global health promotion in general. Firstly, future efforts should investigate the different opportunities that nudges present for targeting NTDs in particular, rather than relying solely on integrated health promotion approaches. Secondly, to apply robust study designs including rigorous process and impact evaluation which allow for a better understanding of 'what works' and 'how it works'. Finally, to consider the ethical implications of implementing nudge strategies, specifically in LMIC.

## Author summary

Behavior is at the core of neglected tropical disease (NTD) prevention and control, certainly within low-, and middle- income countries (LMIC) where resources are often limited. Therefore, strategies to promote behavior change should be included and investigated in future efforts. Nudging, a low-cost strategy that subtly directs people towards positive behavioral choices, has recently gained attention in global health promotion. Nudge strategies have been applied to a wide range of health-promoting behaviors such as handwashing. To understand which strategies were used, where and how these were applied, and whether these were ethically informed and implemented, we undertook a comprehensive review of the available sources. This resulted in 33 included studies, with a total of 67 nudge strategies for behavior-based prevention and control of NTDs in LMIC. Only two studies targeted NTDs in particular, the other 31 included studies were focused on more general health promoting behaviors, with the majority targeting handwashing with soap. The most common nudge strategies were those targeting decision assistance, such as fostering commitment and reminder actions. In general, the ethical assessment presented favorable results. We identified the need for robust study designs to better understand how nudges can be implemented in the future.

## Introduction

In recognition of the underinvestment, global attention and commitment to address neglected tropical diseases (NTDs), the United Nations formally recognized the NTDs, a diverse group of infectious diseases that are common in low-income populations with poor access to quality health services, water, sanitation and hygiene, generally being overlooked and receiving little attention [1]. The Sustainable Development Goals (SDGs) represent global commitments to end the epidemics of AIDS, tuberculosis, malaria and NTDs and combat hepatitis, waterborne diseases and other communicable diseases by 2030 (SDG 3.3) [2]. Currently, preventive chemotherapy, i.e. low-cost mass drug administration without individual diagnosis, is at the core of most NTD programs [3]. However, mass drug administration relies heavily on stable health care infrastructure, which is not always available [4], and additionally, high selection pressure could result in drug resistance and eventually lead to unsustainable control [5]. Therefore, an integrated approach to combating NTDs in the long term is widely

recommended. This implies shifting the focus away from reliance on medicines to integrated and multisectoral approaches [6]. The BEST (Behavior, Environment, Social inclusion, and Treatment) framework was developed by the NTD Non-Governmental Organization Network (NNN) to allow for a comprehensive approach towards NTDs [7]. Nevertheless, comprehensive and integrated approaches, although more effective in achieving sustained control, tend to be more resource demanding [8]. However, the lack of NTD research reflects a pervasive inequality in global health financing [4]. Therefore, there is an urgent need for effective, low-cost interventions targeting NTDs stretching beyond the current therapeutic focus.

Recent studies on handwashing behavior for NTD prevention and control present evidence for "nudging" or similar alternative approaches as effective and low-cost behavior change strategies [9]. The purpose of nudging is to subtly direct people towards positive behavioral choices. Nudging does not preclude people's ability to choose, but instead subtly steers people to make certain favorable decisions for themselves or others. Favorable decisions or favorable behavior relating to health should have a positive health outcome for the people or patients involved [10]. Nudging is grounded in behavioral economics, which is a discipline combining both economics and psychology, and aims to provide an alternative perspective to the assumption that behavior is governed by rational decision-making, as exemplified in traditional economics [11]. The theory acknowledges the limitations inherent to human decision-making and identifies the cause as our "bounded rationality"; i.e., the limitation of human rationality by several factors such as cognitive and emotional biases, peer and time pressure, among other factors [12]. Thaler and Sunstein [13] address these limitations by conceptualizing a nudge as a behavior change strategy, making the insights from behavioral economics more applicable and accessible. According to the authors, who do not offer a definitive definition but merely suggest an interpretation of the term, a nudge is ". . .any aspect of the choice architecture (i.e., the design of different ways in which choices can be presented) that alters people's behavior in a predictable way without forbidding any options or significantly changing their economic incentives. To count as a mere nudge, the intervention must be easy and cheap to avoid" [13] (p.6). The definition of a nudge has since been updated to provide further conceptual clarity and to align with its theoretical underpinnings in the behavioral sciences [14]. Moreover, Thaler and Sunstein's definition of a nudge is difficult to operationalize, as it states only that nudges lead to predictable change in human behavior and are different from significant economic incentives or regulation.

In this scoping review, we use the updated definition by Hansen [15] (p.174), as suggested by O'Keeffe et al. [16] in their review protocol: "A nudge is a function of an attempt at influencing people's judgment choice or behavior in a predictable way, that is (1) made possible because of cognitive boundaries, biases, routines, and habits in individual and social decision-making posing barriers for people to perform rationally in their own self-declared interest, and which (2) works by making use of those boundaries, biases, routines, and habits as integral parts of such attempts. Thus, a nudge amongst other things works independently of: (i) forbidding or adding any rationally relevant choice options, (ii) changing incentives, whether regarded in terms of time, trouble, social sanctions, economic and so forth, or (iii) the provision of factual information and rational argumentation."

Nudges have been found to be effective in promoting health without limiting people's freedom [17], nevertheless, the approach has been widely criticized due to its paternalistic nature. A nudge assumes individuals are not rational actors, capable of making more favorable decisions, which defers responsibility to experts and those in a position of power [18]. Moreover, it questions an individual's autonomy of choice [19]. However, Thaler and Sunstein [20] use the term "libertarian paternalism," which underlines the freedom of choice, whilst attributing some responsibility to the nudger (i.e., the person or group instigating the nudge). Considering

nudge interventions for targeting NTDs, which mainly affect low- and middle- income countries (LMIC) [21], and largely concentrated among the poorest populations [22], it is imperative to be aware of these ethical implications. Nevertheless, even if power dynamics are at play, some researchers advocate for the implementation of nudges, as long as these are suitably transparent and democratically controlled [23].

To date, systematic reviews of nudge interventions designed to improve individual health behaviors have focused on primary preventive behaviors for non-communicable disease control, such as healthy food choices, reducing alcohol consumption, smoking cessation, increasing physical activity, and self-management of chronic diseases [24–30]. To our knowledge, studies focusing on nudge strategies aimed at infectious disease control, and more specifically prevention and control of NTDs, have not been synthesized and this gap presents an opportunity for a scoping review.

The present review aims to provide an overview of how nudge strategies have been applied within health promotion research, with a specific focus on the prevention and control of NTDs, and what the results have shown thus far. Additionally, in order to inform future efforts directed at implementing nudge interventions, we present a set of ethical criteria that can guide the development of future health promotion strategies. In line with these aims and given the relatively recent inclusion of nudges targeting infectious diseases, we opted for a scoping review methodology. This scoping approach allows us to summarize and map the available evidence in nudges for NTD-related research, and additionally, to evaluate relevant ethical considerations that should be taken into account when implementing nudges. The objectives of this scoping review are to: (1) map existing studies which apply a nudge strategy within an intervention for the prevention and control of NTDs in LMIC, (2) identify knowledge gaps to inform future research, (3) propose a framework for the ethical consideration of nudges for NTD prevention and control.

## Methods

In order to ensure a transparent and systematic approach we utilized the JBI Reviewer's Manual methodology for scoping reviews [31], and the Preferred Reporting Items for Systematic reviews and Meta-Analyses extension for Scoping Reviews (PRISMA-ScR) checklist [32] for reporting. We did not aim to systematically assess the quality of the available studies as required for systematic literature reviews, nor were we concerned with the effectiveness of the nudge strategies. We have focused on providing a broad overview of the field of nudging for the prevention and control of NTDs, implemented in LMICs. Due to the interdisciplinary nature of the topic and the relatively recent emergence of nudging as a strategy, a broader approach was preferred over a systematic quality assessment. In that sense, scoping reviews are particularly useful since they bring together literature from diverse disciplines, and with different approaches to health, intervention, and measurement outcomes. To date there have not been any systematic reviews of any kind in the peer reviewed literature which have focused on nudge strategies for infectious disease control, and more specifically prevention and control of NTDs [33].

### Search strategy

Recommendations for scoping reviews suggest that the search strategy be as inclusive and comprehensive as possible. Since the term "nudging" was only recently established and operationalized, studies with a similar focus could potentially be excluded from the review due to different terms used to describe the nudge strategy. The challenge of missing eligible studies because of inconsistent labeling of the term "nudge" has been reported previously [34].

Therefore, we included relevant and alternative terms of nudging, as well as sub-categories of the strategy. To identify and include these terms, we used a similar approach as Möllenkamp et al. [27]. In addition, we aimed to capture all nudge strategies related to prevention and control of NTDs, and not merely targeting these diseases specifically. Therefore, nudge strategies aimed at changing individual or general health-specific behaviors (e.g. handwashing behavior or non-NTD vector control), although not specifically focused on preventing or controlling a particular type or group of NTDs, were also included in the search strategy. The search algorithm is presented in the study protocol [33], and the final search strategy in MEDLINE presented in S1 Text.

A systematic search in the following databases was performed: MEDLINE, PsycINFO, Embase (Ovid), Web of Science Core Collection, CINAHL, ERIC and Econ.Lit (EBSCO), as well as registered trials and reviews in CENTRAL and PROSPERO to identify relevant in-progress or unpublished studies. Moreover, references of included studies and literature reviews were screened (based on "Criteria and screening procedure" below), as well as a citation tracking in Web of Science and Google Scholar. In addition, we searched for gray literature on websites of governmental nudge units, such as the Behavioral Insights Team (UK), the Social and Behavioral Science Team (US), the Behavioral Economics Team (AU), the Ministry of Manpower (SG), and iNudgeyou (DK), but also the World Health Organization (WHO) and WHO trial repository, the World Bank (WB) and WB open knowledge repository, United States Agency for International Development (USAID), the Organization for Economic Co-operation and Development (OECD) and OECD library, the United Nations, Abdul Latif Jameel Poverty Action Lab (Jpal), BVA global and BVA Nudge Unit, Busara Center for Behavioral Economics, The International Initiative for Impact Evaluation (3ie), globalhandwashing. org, and behaviouralscientist.org.

## Selection criteria and screening procedure

Articles were screened based on pre-defined selection criteria and procedures [33]. However, the criteria were adapted due to the nature of the literature and deviated to some extent from the protocol, which is described throughout the methods section. The final eligibility criteria are presented in Table 1. We included all full-text papers and reports, peer reviewed articles and gray literature in English. Initially, no restrictions were placed on the type of evidence or study design. However, these inclusion criteria were adapted fairly rapidly through the process, since an outcome measurement of the intervention was required for the data extraction. Review articles were excluded but screened for identification of missed studies that were

**Table 1. Final eligibility criteria.**

| Inclusion criteria | Exclusion criteria |
|---|---|
| Full-text papers and reports, peer reviewed articles and gray literature | Literature reviews, conference abstracts, editorial letters and comments, theoretical/background papers |
| Written in English language | Written in languages other than English |
| Context of the study is low- and middle- income countries | Context of the study is high-income countries |
| Targeted all behavioral practices leading to NTD prevention and control | Targeted other types of behavioral practices leading to prevention and control of other diseases |
| Behaviorally informed intervention strategies attributed to a nudge strategy, regardless of this explicit label | Interventions not attributed to a nudge strategy, according to the definition by Hansen (2016) |
| The nudge strategy described in full | An incomplete description of the nudge strategy |
| Study design makes it possible to isolate the effect of the nudge on health/implementation outcomes | Study design does not allow for the measurement of the effect of the nudge |

subsequently added to the database of included studies. Conference abstracts, as well as editorial letters and comments, were excluded from further review. Additionally, to be eligible for inclusion, the nudge strategy had to be described in full, excluding all studies that had an incomplete description of the intervention. All interventions including a nudge strategy were reviewed, regardless of whether they included the explicit label of a "nudge" or not. Initially, to count as a nudge we considered the description by Thaler and Sunstein [13], however, this quickly resulted in an excessive number of studies. Thaler and Sunstein's definition of a nudge is difficult to operationalize, as it states only that nudges lead to predictable change in human behavior and are different from significant economic incentives or regulation. Therefore, we consulted an expert in nudging, and adopted the more delimited definition of nudging proposed by Hansen [15]. Thus, articles describing a type of intervention strategy that did not qualify as a nudge based on the definition by Hansen [15] were excluded. Finally, no adaptations were made to the population and context criteria, which included all populations exposed to a nudge strategy targeting behavioral practices for NTD prevention and control in LMICs. All contexts were considered eligible, e.g. public spaces, health care facilities, school settings, and indoor/outdoor community facilities.

All literature was downloaded by one review author (FVV) to EndNote X9 (Clarivate Analytics, PA, USA), and duplicates removed. The titles and abstracts were screened by one reviewer only (FVV), due to the large number of results. Eligible studies were selected through a questionnaire that specified the inclusion criteria. Two reviewers (SB and FVV) subsequently reviewed the full-text studies independently from each other. Any disagreements concerning eligibility were resolved through discussion. In addition, two independent experts in nudging were contacted to resolve disagreement to eventually reach a consensus.

## Data extraction

All papers selected for inclusion were subjected to a data extraction procedure designed by one review author (FVV) and agreed upon by the other review authors. The extraction form included the following predefined categories: authors; journal/source; year of publication; type of publication; geographical area of the intervention; setting of the intervention; domain of preventive practices (e.g. hygiene, vector control); targeted behavior description (e.g. handwashing); targeted NTD, if specified (e.g. dengue); targeted population; protection of self or others (i.e. whether the behavior is focused on protecting oneself such as through handwashing, or other behaviors such as vector control in the community); nudge strategy; underlying theory; and intervention results. Moreover, the nudge strategies were categorized using a choice architecture taxonomy developed by Münscher et al. [35] (Table 2), similar to Forberger et al.'s scoping review on nudges to promote physical activity [25].

In addition, given that nudges raise ethical concerns, the interventions were evaluated based on an adapted ethical framework developed by Engelen [36]. To bridge the gap between the abstract theoretical debates among academics and the actual behavioral interventions being implemented in practice, Engelen developed a set of criteria to facilitate an ethical assessment of different types of nudges [36]. The corresponding categories, criteria and coding labels were adapted to fit the scope of the review, as well as the scope of the included studies, and are presented in Table 3. The original framework presented nine ethical criteria that are categorized into three larger groups: 1) criteria for ends, i.e. evaluation of people's goals and values; 2) criteria for means, i.e. evaluation of people's decision-making process; 3) criteria for agents i.e. evaluation of the role of the nudgers. In the current scoping review, the categories are reflected accordingly: 1) criteria for targeted behaviors, which are considered the 'ends' or the

**Table 2. Choice architecture categories and strategies by Münscher et al., 2016.**

| Category | Description | Strategy | Examples |
|---|---|---|---|
| A. Decision information | Presenting decision-relevant information without changing existing options | A1 Translate information | Message reframing, simplifying |
| | | A2 Make information visible | Feedback on own behavior, accessibility of external information |
| | | A3 Provide social reference point | Refer to a descriptive norm, or an opinion leader |
| B. Decision structure | Designing or changing options and associated consequences | B1 Change choice defaults | No-action default, prompted choice |
| | | B2 Change option related effort | Increase/decrease of physical/financial effort |
| | | B3 Change range or composition of options | Change category/group of options |
| | | B4 Change option consequences | Change benefit/cost/social consequences of the decision |
| C. Decision assistance | Supporting existing intentions to change | C1 Provide reminders | Un-/materialized reminders |
| | | C2 Facilitate commitment | Support self- or public commitment |

result of the implemented strategy; 2) criteria for interventions, which relate to the nudge strategy and process in itself; and 3) criteria for researchers, the agents responsible for implementing the study.

The ethical criteria were subsequently adapted to ensure relevance and facilitate the assessment on the included studies. We aimed at adhering to the original criteria, but this was not always possible due to the limited information reported in the manuscripts. Three criteria changed categories: low processing motivation, democratic legitimation, and easy resistibility. For example, 'low processing motivation' became a criterion for the targeted behaviors, since we were not able to deduce the way the intervention (e.g., the means) was processed by the participants, based on the included information. However, the studies included information on the repetitiveness and importance of the targeted behavior (e.g., habitual handwashing, taking care of infants), hence the necessity of performing the behavior through a high- or low-cognitive route. Extracting this information allowed us to develop coding labels, and subsequently evaluate the criterion. A similar process supported the development of all other criteria. The criterion 'rational capacities' was excluded from the assessment, since we were unable to identify relevant coding labels that fit all the studies. Finally, we developed three assessment codes to indicate the degree to which studies met each criteria: High (H); Moderate (M); Low (L).

The ethical assessment was developed concurrently with the data extraction form and both were pilot tested to minimize misinterpretation and to ensure all relevant data were included in the analysis. Given the diversity of studies, the piloting was conducted among a sample of five studies comprising different domains of health promotion practices and with diverse study designs. The piloting was performed by one review author through several iterations (FVV) and discussed at length with a second reviewer (SB). This resulted in the inclusion of several categories in the data extraction form: authors' affiliations; behavioral intervention package complexity (i.e. largest number of intervention components implemented to a group participants; simple: 1–2, moderate: 3–4, and complex: >4); theoretical underpinning or design process of the whole intervention package; nudge materials (e.g., posters, painted cues, messages); research design; secondary outcomes related to the nudge; intervention outcomes attributed to the nudge; experimental design. Ultimately, the data extraction procedure was conducted by one review author (FVV), whilst a second reviewer (SB) validated 25% of the extracted information. Disagreements were resolved through discussion until consensus was reached.

**Table 3. Criteria adapted from Engelen (2019) for ethical assessment of nudge strategies targeting health promotion behaviors related to neglected tropical diseases in low- and middle-income countries.**

| Category | Criteria | Description | Label | Code[1] |
|---|---|---|---|---|
| 1. Criteria for targeted behaviors | Reflective preferences | Nudges should be based on people's own reflective preferences | Targeted behavior is underpinned by population preferences | H |
| | | | Formative research conducted, but not used to target the behavior | M |
| | | | No mention of formative research or behavioral preferences | L |
| | Health benefits | Nudges should generate improved health outcomes | Health benefits are immediate by implementing the behavior | H |
| | | | Health benefits depend on the involvement of the community | M |
| | | | Health benefits depend on many other environmental factors | L |
| | Low processing motivation | Nudges should require low processing motivation | Behavior is repetitive, low processing is needed | H |
| | | | Behavior is implemented during certain moments, or for a certain group | M |
| | | | Behavior is performed only once, therefore high stakes | L |
| 2. Criteria for interventions | Democratic legitimation | Nudges should be based on broad public support | Intervention is developed through participatory approaches or iterations | H |
| | | | Intervention is with insights from formative research or by a local agency | M |
| | | | Intervention is not developed with community reflection or feedback | L |
| | Easy resistibility | Nudges should allow people with opposite preferences to go against them | Awareness of the nudge and other options are within reach | H |
| | | | Awareness of the nudge, but other options are fairly unreachable | M |
| | | | No awareness of the nudge and no other options available | L |
| | Long-run autonomy | Nudges should generate greater autonomy in the long run | Both short- and long- run autonomy are preserved | H |
| | | | No short-, but long- run autonomy is preserved | M |
| | | | Both short- and long- run autonomy are not preserved | L |
| | Available alternatives | Nudges should be more effective than information or persuasion | Nudge showed a positive outcome, more effective than alternatives | H |
| | | | Inadequate experimental design, but study showed a positive outcome | M |
| | | | Nudge resulted in a negative outcome, less effective than alternatives | L |
| 3. Criteria for researchers | Trust relationship | Nudges should be implemented by agents in a trust relationship with the nudgees | One of the authors affiliated with a local institution | H |
| | | | No authors affiliated with a local institution, but includes local collaborators | M |
| | | | No mention of local collaboration throughout the study | L |

[1]H = high ethical standard; M = moderate ethical standard; L = low ethical standard

## Results

### Study selection

The search strategy resulted in a total of 2497 records retrieved from the specified databases, with 1792 records remaining after removal of duplicates (Fig 1). Similar to a method used by

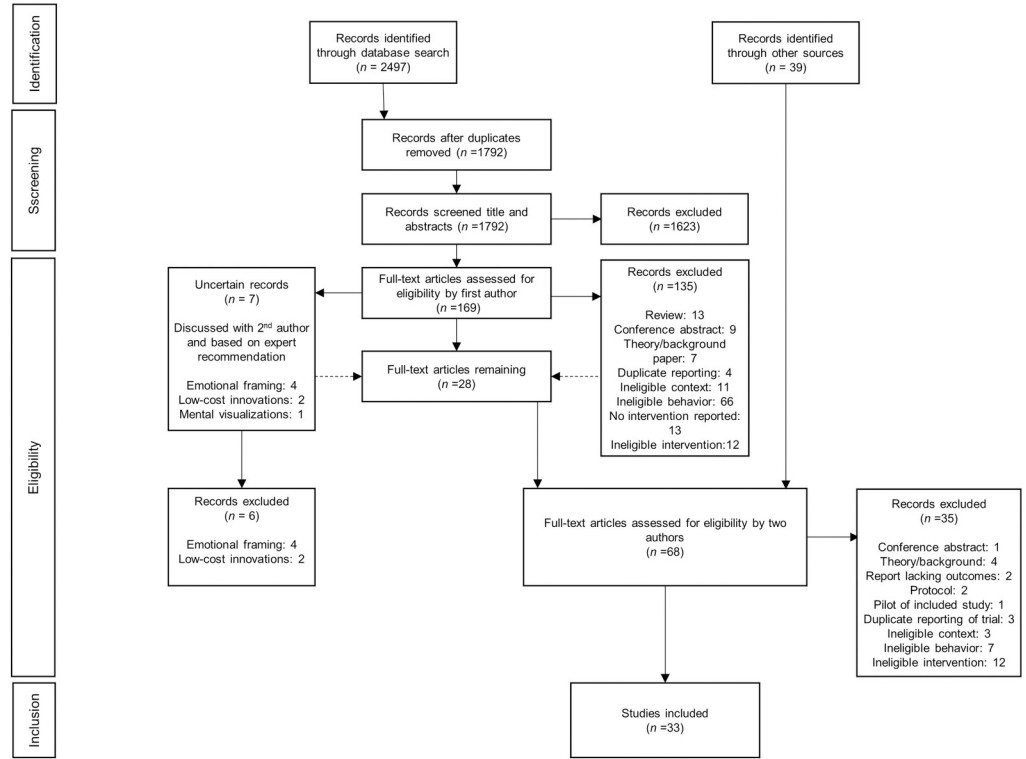

**Fig 1. Flow diagram based on the Preferred Reporting Items for Systematic reviews and Meta-Analyses extension for Scoping Reviews (PRISMA-ScR) checklist (Tricco et al., 2018).** Note that the selection process was iterative and resulted in reconsideration and subsequent inclusion of studies for eligibility assessment (dashed lines).

Weston et al. [37], the selection process was iterative as the criteria were developed based on the discussions amongst the review authors and experts' inputs and became more exclusive and bounded to improve conceptual clarity. In the first full-text examination round, one review author (FVV) sorted the 169 selected records into 'include', 'exclude' or 'uncertain' selection. The 'uncertain' selection was discussed with a second review author (SB), and eventually synthesized into three main topics of uncertainty: emotional framing, low-cost innovations, and mental visualizations. We reached out to two experts to assist with resolving the uncertainty associated with these approaches. Additionally, one expert provided additional premises and guidelines: a nudge is an aspect of an intentional intervention, not the intervention itself; the intervention can therefore include both rational aspects as well as nudges at the same time; a nudge does not affect action by provision of rational reasoning; actions can also include mental events (e.g. belief formation, direction of attention); simply creating an opportunity does not count as a nudge; fear of social sanctions does not count as a nudge; and lastly, a nudge implemented to alter the mental or emotional state should be supported by evidence and not merely assumed. Subsequently, the eligibility criteria were modified, uncertain records were examined for inclusion, and previously excluded records based on 'ineligible intervention' were re-assessed. After an extensive search for potentially missed studies and gray literature, concluded on September 6[th], 2020, two independent reviewers (SB and FVV) initiated a second round of full-text examination of a total of 68 records. Eventually, and with far less discussion, this resulted in 33 identified studies.

## Notable eligibility decisions

Despite the application of stringent eligibility criteria in an objective and systematic fashion, there is unavoidably an element of subjectivity involved, and certainly when considering the complexity of the nudge definition. In an attempt to make this subjectivity transparent, this sub-section reports on the eligibility decisions made. These decisions led to the selection for inclusion of a full study, or the selection of an intervention, if the study implemented multiple nudge strategies within the intervention.

According to the eligibility criteria, the nudge intervention had to be described in full, excluding all studies that had an incomplete description of the intervention. This criterion is closely associated with the expert's premise that a nudge is an aspect of an intentional intervention, and not the intervention itself. Due to this premise, we excluded a number of studies that implemented a 'reminder' as an aspect of the intervention, without explicitly describing the process the reminder was targeting. Merely describing the instrument (e.g., poster, song) was not sufficient to be included in the analysis. Some studies implemented posters or murals as reminders to instigate a certain preventive behavior, but failed to describe the precise location for this reminder to be activated (e.g. posters near the sink to activate handwashing action). Interventions that mentioned 'posters were placed in a strategic location' [38] or even developed and implemented through participatory processes [39] were further excluded from analysis if they lacked sufficient detail to the context and location of the nudge. Interventions implementing songs or radio advertisements as reminders were also excluded from the analysis, since the timing of these interventions was not controlled and therefore it was not possible to be considered as a reminder [40].

Targeting of social norms was applied in many interventions and studies, however, such an approach must be used in a certain manner in order to be labeled a nudge. The norms should be descriptive (e.g., what people do), rather than injunctive (e.g., what people should do), since the latter induces fear of social sanctions. Subsequently, avoiding social sanctions is regarded as a rational decision and is, therefore, not a nudge. Any form of peer pressure or promotion of injunctive norms were excluded from further analysis. For example, an intervention on infant feeding in Indonesia based on gossip induced fear of social sanctions among young mothers if they did not implement adequate infant feeding practices [41]. Studies were also excluded on the basis of inadequate research design. If the study design did not allow for deducing the effect of the social norms, due to missing secondary outcomes or insufficient evidence of the intervention being linked to social constructs, the intervention was deemed ineligible [42].

Similar to the latter argument presented for social norms, we excluded interventions that aimed at prompting action by altering decision information through certain cognitive or emotional processes without explicitly measuring this hypothesized cognitive or emotional model. Several interventions based on emotional triggers such as disgust and nurture were not included due to a lack of empirical information about the intervention processing [43,44]. We were not able to infer whether the action was instigated by merely a matter of information provision or whether participants were exposed to a nudge. So called 'triggering' motivating behavior change by activating a collective sense of disgust [45], mainly used in community-led total sanitation efforts, were equally subjected to this exclusion argument.

The provision of tippy taps, i.e. locally made low-cost devices for washing hands with running water [46] or other 'enabling technologies', were not considered eligible interventions. The majority of these interventions merely provided an opportunity, which eventually created the desired action (i.e. handwashing). In addition, these interventions can arguably be categorized as some form of frugal and innovative sanitary engineering, which also falls beyond the scope of this review.

## Study characteristics

As detailed previously, 33 papers were retained following full-text analysis [47–79]. This section will present the eligible records and the extracted data relating to the study characteristics. The complete list of citations and extracted data presented below are available in S1 Data. The full extracted data is available on request from the first author.

**Date of publication.** All included records were published since 2010 with the number of studies increasing exponentially, with a substantial rise from 2017 onwards. During that first modest increase (2010–2017), 12 studies were published [48,50,55,58,65,68–70,72–74,76]. The exponential increase is most noticeable in 2018 resulting in 7 eligible studies [51,52,57,59,64,66,67], while in 2019, there were 9 included studies [47,53,54,61,63,71,75,78,79], and finally up until 6th September 2020 we identified 5 additional studies [49,56,60,62,77].

**Study locations.** Half of the included studies were conducted in Asia (16 records), with India making up the largest cohort of studies [48,53,56,57,66,72,77], followed by Bangladesh [47,50,59,68], the Philippines [60,63] Nepal [58], and Iraq [78]. One study was conducted in both Bangladesh and Kenya [69]. The other half of the included studies were conducted in Africa (15 records): four studies in Kenya [61,64,75,79], followed by two in Zimbabwe [62,65], Nigeria [49], South Africa [51], Malawi [54], Ethiopia [55], Chad [67], Burundi [70], Zambia [71], Mali [73], Egypt [74], and Uganda [76]. Finally, only one study was identified in South America, in Peru [52].

**Setting and participants.** Half of the studies were conducted in a rural setting (16 records [48–50,53–58,61,62,64,67,70,73,77]). Eight studies took place in an urban/peri-urban setting, of which six studies specified a low-income/slum context [47,51,68,69,72,76], the remaining two did not specify the context of the setting [52,65]. A total of eight studies specifically targeted schools [59,60,63,66,71,74,75,79], and one study focused on a humanitarian emergency setting [78].

The participants included only mothers or caregivers of children under the age of 5 (5 records [47,55,56,58,67]), both caregivers and children (2 records [66,72]), and only including children (8 records [51,59,60,63,71,75,78,79]). Several studies targeted an entire household (8 records [48–50,52,56,57,69,77]) or the entire community (7 records [53,62,64,65,68,70,73]), whilst fewer studies specified targeting the school community [74], young women [61], or users of a certain facility with specified traits such as shared compound toilets [76].

**Targeted behavior.** The included studies did not specify a disease which the intervention was focused on specifically, apart from two studies targeting Chagas disease [52] and trachoma [75]. The studies targeted specific behaviors rather than diseases and merely described possible health effects of these behaviors, mostly a decrease of infectious pathogens causing a host of illnesses (e.g., diarrhea, pneumonia). Half of the studies targeted handwashing with soap (HWWS) or simply handwashing (17 records [47–51,55,59,60,62,63,66,68,71,72,74,78,79]). Seven studies targeted defecation behaviors such as latrine usage, building or cleaning, and safe disposal of child feces [53,56,57,70,73,76,77], four studies focused on water disinfection through chlorination or solar disinfection [61,65,67,69], two studies targeted a mix of food hygiene behaviors [54,58], and one study each targeted drug intake for deworming [64], face washing [75], and engaging the population to participate in an indoor residual spraying (IRS) campaign [52].

**Intervention strategies.** A range of intervention strategies were employed across the included studies, with some studies detailing multiple, combined behavior change strategies (including nudges), others incorporating few strategies, whilst some focused on applying one strategy. Half of the eligible studies consisted of a complex intervention, including more than

four behavior change strategies (17 records [48,49,53,54,56,58,62,63,65–68,72–75,77]). Subsequently, a total of seven 'moderate intervention' studies were recorded that included 3–4 behavior change strategies [50,52,55,57,60,69,79]. Finally, nine records consisted of a simple intervention, including up to two behavior change strategies [47,51,59,61,64,70,71,76,78].

**Study design.** The included studies were based on a range of different research designs, with the majority implementing some type of randomized controlled trial (22 records [47–49,52,53,56,58–64,66,68,69,71–75,79]). Six studies implemented a longitudinal design with baseline and end-line measurements [54,55,65], three of them included an intervention and a control group [70,76,77]. Four pilot studies were included, of which two performed baseline and end-line measurements [57,78], one consisted of a randomized controlled design [51] and another implemented an experimental design that compared two different intervention groups [50]. Finally, one study reported a cross-sectional design with intervention and control group [67].

The majority of the study designs were not able to isolate and measure the impact of separate behavior change strategies included in the intervention strategy (22 records [48–50,52–54,56,58,60,62,63,65–68,70,72–75,77,79]). The remaining 11 studies were able to measure the impact of distinct behavior change strategies on the targeted behavioral outcomes [47,51,55,57,59,61,64,69,71,76,78].

## Nudge characteristics

We identified a total of 67 nudges across 33 studies. This section presents the extracted nudges based on the targeted behavior, the different categories of choice architecture, and ethical standards.

**Nudges and targeted behavior.** Thirty of the nudges were implemented for HWWS in a total of 17 studies [47–51,55,59,60,62,63,66,68,71,72,74,78,79] (Fig 2). This group of nudges

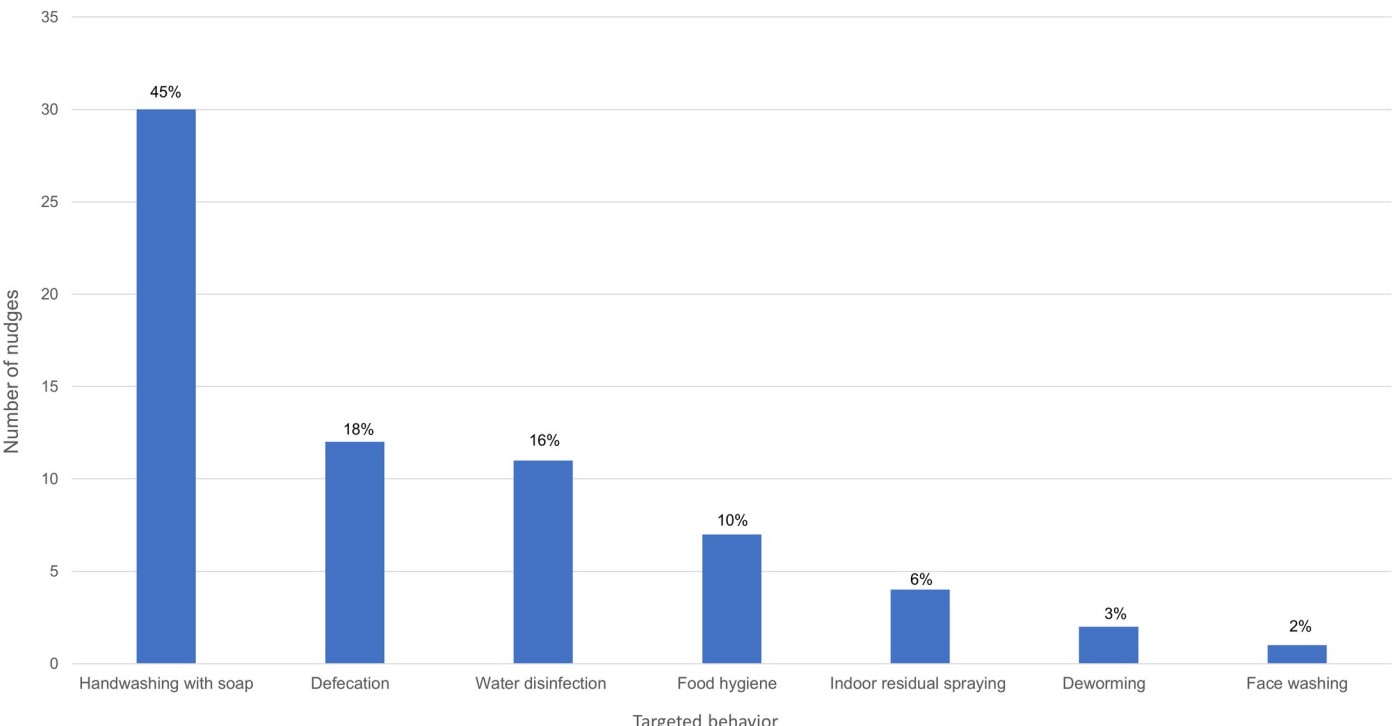

**Fig 2.** Number of nudges included in the review (n = 67), targeting different behaviors: Handwashing with soap; Defecation behaviors; Water disinfection; Food hygiene; Indoor residual spraying; Deworming; Face wahing.

were most diverse and included a range of behavior change strategies, such as the implementation of disruptive cues: colorful, painted footpaths [59,60]; soap including embedded toys [51,78]; soap on a rope [71]; posters as reminders [60,68,74]; stickers as reminders [48,50,60], targeting social norms: stickers or certificates as signals of pro-social behavior [48, 49, 68]; role-model display [48]; and public pledges [48,49,55,62,63,66,72,79] by simplifying information [47] and providing planning reminders [62]. Twelve nudges targeted defecation behaviors in 7 studies [53,56,57,70,73,76,77]. Most of these studies deployed commitment as a strategy, e.g. self-commitment [53,70] or public pledges [56,73,76,77]. Others included strategies targeting social norms such as signaling (banners [53], family picture [56]); role models [57]; public competitions [73], but also by providing sticker reminders [56], and by providing visual feedback in coloring the village open defecation "hot spots" [53]. Disinfecting water was targeted by 11 nudges in 4 studies [61,65,67,69]: signaling (stickers [65] or ribbons [69]), public pledges [67,69], message framing [67,69], reminder stickers [65], simplifying the behavior through planning [61], or visualizing the future [61]. Only two studies focused on food hygiene behaviors, but these included a total of 7 nudges: public pledges [54,58], signaling (stickers [54] and 'safe food' zones [58]), competitions, colorful reminders in the kitchen and role-model display [58]. One study targeted vector control and participation in an IRS campaign for Chagas' disease control [52], including 4 nudges: planning commitment, associated planning-reminders, role-models promoting participation, and contingent group lotteries. Another study included two nudges targeting deworming behavior using social signaling through bracelets and reminder messages [64]. Finally, one nudge targeted face washing through a public pledge in a classroom setting [66].

**Choice architecture.** The included nudge strategies were categorized into different strategies (Table 2) and depicted in Fig 3. The most common nudge strategies were those targeting decision assistance, such as "facilitating commitment" (24 nudges), representing both self-commitment (including planning commitment) and public commitment (public pledges) [48,49,52–56,58,62,63,66,67,69,70,72,73,75–77,79] and "providing reminders" (16 nudges), using posters, stickers, and planning reminders [48,50,52,56,58–60,62,64–66,68,71,74]. Strategies targeting decision structure represented by "change option consequences" only, included 16 nudges using signaling, competitions and lotteries (including toy-soap, since the chances of 'winning' the toy depends on the amount of soap used) [48,49,51–54,56,58,64,65,67,68,73,78]. Finally, decision information was the least used category with 5 nudges using "translate information" through simplification and message reframing [47,61,67,69], 3 nudges using "provide social reference point" referring to role-models [48,52,57,58], and 2 nudges implementing "make information visible" through visualization of the future and providing visual feedback [53,61].

**Ethical assessment.** The results of each of the eight ethical criteria described in Table 3 are presented below, one by one, and visualized in Fig 4 through the corresponding ethical standard (i.e., H = high; M = moderate; L = low). An overview of the applied framework and a complete list of ethical coding (H, M and L) and corresponding citations can be found in Table 4.

"Reflective preferences"–RP were assessed by extracting the data regarding formative research; High, 8 nudges were developed through behavioral preferences of the targeted population (e.g. identifying irrelevant behavior to target together with the community); Moderate, 39 nudges were developed through pre-defined behavior, but with some formative knowledge of the population (e.g. assessing the barriers or norms to a certain behavior); Low, 20 nudges were developed without any research on the preferences of the studied population (e.g. no mention of formative research).

"Health benefits"–HB were assessed based on whether the targeted behavior resulted in immediate positive health outcomes or to what degree these health outcomes depended on other factors apart from the behavior: High, 51 nudges had immediate health benefits (e.g.

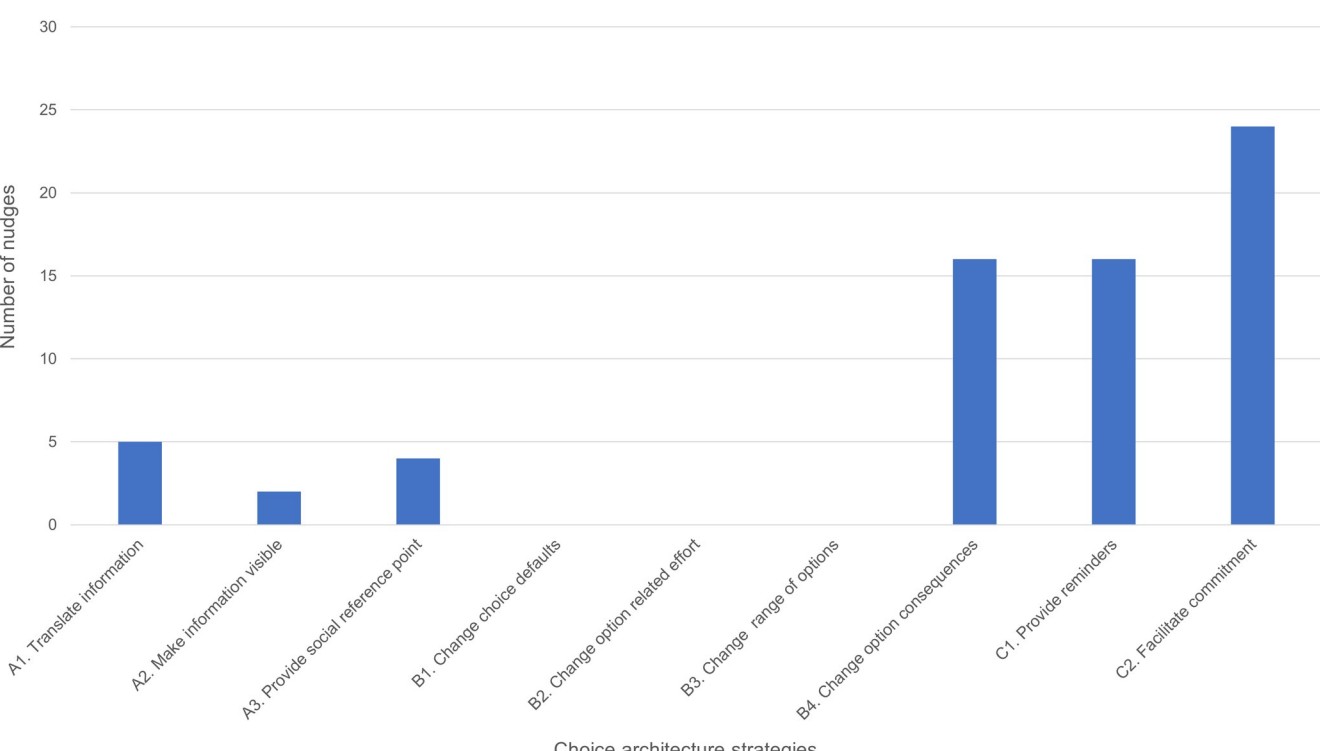

**Fig 3.** Number of identified nudge strategies in selected studies based on the three choice architecture categories A. Decision information, B. Decision structure, and C. Decision assistance (Münscher et al. 2016). A1 Translate information; A2 Make information visible; A3 Provide social reference point; B1 Change choice defaults; B2 Change option related effort; B3 Change range or composition of options; B4 Change option consequences; C1 Provide reminders; C2 Facilitate commitment.

HWWS); Moderate, 10 nudges depended on community participation (e.g. defecation behaviors); Low, 6 nudges depended on other environmental factors, beyond the targeted behavior (e.g. cleaning latrines).

"Low-processing motivation"–LP was assessed through the repetitiveness of the targeted behavior, as well as the magnitude of the impact: High, 23 nudges targeted low-processing behaviors (e.g. HWWS); Moderate, 38 nudges targeted behaviors during specified moments such as water disinfection, or involving a certain group (e.g. infant feeding); Low, 6 nudges were targeting one-time, key decisions (e.g. deworming).

"Democratic legitimation"–DL was assessed through extraction of the data regarding the development of the intervention: High, 34 nudges were developed with the community (e.g. co-creation of intervention materials); Moderate, 8 nudges were based on formative research, but without direct community involvement (e.g. local design agency); Low, 25 nudges were designed by the research team without local involvement or knowledge (e.g. materials were pre-designed).

"Easy resistibility"–ER was based on the awareness of the nudge and whether other options (or opting out) are visible and within reach of the nudgee: High, 28 nudges were highly visible and easy to avoid if the participant preferred to not engage (e.g. reminders); Moderate, 39 nudges were also visible, but more difficult to oppose (e.g. all public behaviors); Low, no nudges were invisible or hiding other options (e.g. default).

"Long-run autonomy"–LA was based on the assessment of both short-term decisions and modifications during the intervention, and longer-term changes after the study has come to a

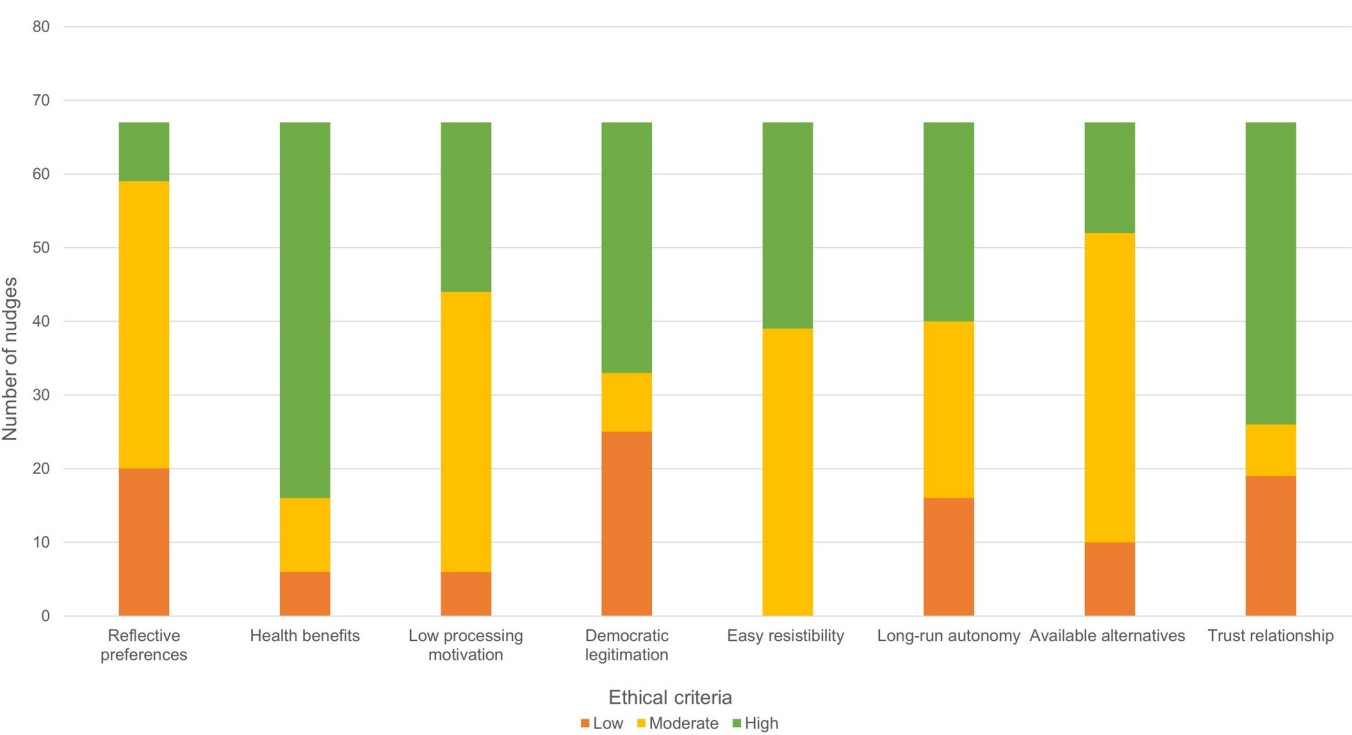

**Fig 4. Ethical assessment of 67 nudge strategies targeting health promotion behaviors related to neglected tropical diseases in low- and middle-income countries.** Nudge strategies were assessed according to eight criteria: Reflective preferences (RP); Health benefits (HB); Low processing motivation (LP); Democratic legitimation (DL); Easy resistibility (ER); Long-run autonomy (LA); Available alternatives (AA); Trust relationship (TR), and coded into high (H), moderate (M), and low (L) ethical standard.

conclusion: High, 27 nudges secure both short- and long- run autonomy (e.g. reminders); Moderate, 24 nudges affected short-term decisions, but long-run autonomy is preserved (e.g. public commitment); Low, 16 nudges make significant changes to the environment and result in both short- and long- run loss of autonomy (e.g. signaling).

"Available alternatives"–AA were assessed through the reported health outcomes of the studies: High, 15 nudges were more effective than the presented alternative options (e.g. adequate experimental design with a positive effect attributed to the nudge); Moderate, 42 nudges were part of a larger intervention set that reported a positive health outcome (e.g. intervention set compared to other or control); Low, 10 nudges reported negative effects compared to the alternative (e.g. adequate experimental design with a negative effect attributed to the nudge).

Finally, "trust relationship"–TR consisted of one item assessing the trust relationship between the nudgers and the nudgees, in this case the relationship between the research team and targeted population: High, 41 nudges were implemented by a local collaborator as part of the research team (e.g. minimum of one affiliated author); Moderate, 7 nudges were implemented by a local collaborator that was not part of the research team (e.g. no locally affiliated author); Low, 19 nudges were implemented by a foreign research team without mentioning local collaboration (e.g. the study did not mention local involvement).

## Discussion

In this scoping review, we identified a total of 33 studies that included a nudge strategy for behavior-based prevention and control of NTDs in LMICs. Only two studies targeted NTDs in particular, trachoma and Chagas disease, the other 31 included studies focused on more

**Table 4. The ethical assessment of the included nudge strategies targeting health promotion behaviors related to neglected tropical diseases in low- and middle-income countries.**

| Studies | Nudges | Reflective preferences | Health benefits | Low-processing motivation | Democratic legitimation | Easy resistibility | Long-run autonomy | Available alternatives | Trust relationship |
|---|---|---|---|---|---|---|---|---|---|
| Amin et al., 2019 | 1. Simplification | L | H | M | L | H | H | L | H |
| Biran el a., 2014 | 1. Public pledges | M | H | M | H | M | M | M | H |
| | 2. Posters role-model | M | H | M | H | H | H | M | H |
| | 3. Prompts of pro-social behavior | M | H | M | H | M | L | M | H |
| | 4. Reminders in bathroom | M | H | M | H | H | H | M | H |
| | 5. HWWS village certificates | M | H | M | H | M | L | M | H |
| Biran et al., 2020 | 1. Public pledges | M | H | H | H | M | M | L | H |
| | 2. Stickers as signals | M | H | H | H | M | L | L | H |
| Biswas et al., 2017 | 1. Reminder sticker | M | H | M | M | H | H | M | H |
| Burns et al., 2018 | 1. Hope soap | L | H | H | L | M | M | H | H |
| Buttenheim et al., 2018 | 1. Advanced Planning | H | L | L | H | H | H | L | H |
| | 2. Reminder planning | H | L | L | H | H | H | L | H |
| | 3. Block Leader Recruitment | H | L | L | H | H | H | L | H |
| | 4. Contingent Group Lotteries | H | L | L | H | M | M | L | H |
| Caruso et al., 2019 | 1. Transect walk with colors | M | M | M | H | H | M | M | H |
| | 2. Banners and mural—signaling | M | M | M | H | M | L | M | H |
| | 3. Self-Commitment and poster | M | M | M | H | H | H | M | H |
| Chidziwisano et al., 2019/2020 | 1. Public pledges | M | H | M | M | M | M | M | H |
| | 2. Stickers as signals | M | H | M | M | M | L | M | H |
| Contzen et al., 2015 | 1. Public-commitment | M | H | M | M | M | M | L | M |
| Friederich et al., 2020 | 1. Public pledge | M | M | M | M | M | M | M | H |
| | 2. Photo as signal | M | M | M | M | M | L | M | H |
| | 3. Stickers as reminders | M | M | M | M | H | H | M | H |
| Gauri, et al 2018 | 1. Norm entrepreneurs | M | M | M | H | H | H | M | L |
| Gautam et al., 2017 | 1. Public pledges | M | H | M | H | M | M | M | M |
| | 2. Competitions | M | H | M | H | M | L | M | M |
| | 3. Reminders in kitchen | M | H | M | H | H | H | M | M |
| | 4. Role-model pictures | M | H | M | H | H | H | M | M |

*(Continued)*

**Table 4.** (Continued)

| Studies | Nudges | Reflective preferences | Health benefits | Low-processing motivation | Democratic legitimation | Easy resistibility | Long-run autonomy | Available alternatives | Trust relationship |
|---|---|---|---|---|---|---|---|---|---|
| | 5. Signaling safe food zones | M | H | M | H | M | L | M | M |
| Grover et al., 2018 | 1. Contextual painted cues | L | H | H | M | H | H | L | H |
| Haung et al., 2020 | 1. Contextual painted cues | L | H | H | H | H | H | H | H |
| | 2. Posters simplifying info | L | H | H | H | H | H | H | H |
| | 3. Eyes sticker | L | H | H | H | H | H | H | H |
| | 4. Arrow sticker | L | H | H | H | H | H | H | H |
| Haushofer et al., 2019 | 1. Visualization | L | H | M | L | H | H | H | H |
| | 2. Planning | L | H | M | L | H | H | H | H |
| Inauen et al., 2019 | 1. Planning reminders | M | H | H | L | H | H | M | L |
| | 2. Public Commitment | M | H | H | L | M | M | M | L |
| Jetha et al., 2019 | 1. Public Pledge | M | H | H | H | M | M | M | H |
| Karing et al., 2018 | 1. Social signaling | H | H | L | H | M | L | H | H |

Note. H = high ethical standard; M = moderate ethical standard; L = low ethical standard

general health promoting behaviors such as handwashing behaviors. Accordingly, handwashing with soap accounted for almost half of the included nudge strategies, followed by defecation behaviors (e.g., latrine usage, cleaning), and water disinfection behaviors. Targeting health behaviors rather than diseases, hence implementing an integrated, horizontal- rather than a vertical approach, has increasingly become common practice in global health and health promotion, and specifically concerning WASH-related efforts [80].

This integrated approach is also reflected in the complexity of the studies and the inclusion of multi-layered, complex interventions. The majority of the studies implemented interventions that included several different behavior change strategies, which made it particularly difficult to identify nudge strategies in reported studies. Consequently, nudges were difficult to isolate, and eventually to attribute their effectiveness within a complex intervention. Although this was not the purpose of our scoping review, it identifies an important knowledge gap that needs addressing in terms of advancing the field. Nevertheless, these studies also provide evidence and support for incorporate nudges in an integrated approach for combatting NTDs, as put forward by the NNN in their BEST framework [7]. Nudges have shown to be low-cost, and context-specific strategies that are relatively easy to include in complex interventions. The findings of this review provide an overview of an increasingly commonly used approach to promoting behavior change through altering people's environment and behavior, by which we aim to facilitate future research in the field. Moreover, although horizontal health promotion approaches have become common practice, the results of this review point to the need for additional research that focuses on nudges in relation to specific health-promoting behaviors for combatting NTDs in particular. Several NTDs are acquired in specific contexts, and avoided by certain behaviors, that are not considered by horizontal health promotion practices: for instance, trachoma is prevented by face washing, schistosomiasis by avoiding contact with contaminated water, snakebite by wearing protective footwear. By altering a specific

environment, nudges present an opportunity to focus on such context- and behavior-specific traits of NTDs. Therefore, future efforts should, in addition, take this specific nature of certain NTDs into account.

Based on Münscher et al. [35] categorization of choice architecture, we found that decision assistance nudges (category C) were most popular in targeting behaviors for NTD prevention and control, followed by nudges targeting decision structure (category B) and finally nudges altering decision information (category A). Category C nudges are most transparent and ethical since they merely provide assistance to help nudgees to follow through with their intentions. Commitment nudges (C2) were also commonly applied, and these can be distinguished between self- and public- commitment. However, we did not differentiate these since most self-commitment nudges to some extent involved public commitments, due to at least one witness as part of the research team being present. Reminders (C1) were frequently reported, however, due to the lack of detailed description concerning the context in which they were implemented, we were unable to consider them as nudges.

Category B interventions are considered least transparent or ethical due to hiding or changing other, less favorable options. Noteworthy is that we only found type B4 nudges, change option consequences (social or beneficial/costs), which is to some extent in line with other reviews focused on nudging and health-related behaviors [25,27]. Distinct from these other reviews, our included studies targeted social consequences/encouragement, rather than connecting these decisions to micro-benefits or costs. One explanation could be that in LMICs, and in particular rural settings, the importance of the community takes greater precedence than the individual. Thereby making this type of nudge more appropriate and strategic to target social consequences/encouragement, such as 'signaling' pro-social behaviors.

Finally, nudges based on category A were reported least frequently. Similar to reminders, many of the studies including message framing (e.g., emotions), targeting of social norms, visualizing or simplifying information were excluded from further analysis, due to a lack of detailed description and validation of the process. Triggering negative emotions such as disgust or shame, or positive emotions such as nurture and fun, are frequently used in global health promotion and specifically in community-led total sanitation approaches [45]. However, there is a significant lack of empirical verification of the causality of these emotional processes. We therefore question the use and effect of many of these interventions, as have others [81], which presents another important research gap.

## Identifying areas for future research

Two major areas for future research were identified through this scoping review, which present an opportunity to learn from and include other research methodologies and fields: (1) 'what works'; and (2) 'how it works'. Firstly, the majority of the included studies consisted of a complex set of interventions, which is common practice in health promotion and is largely encouraged [82]. However, many of these studies were not concerned with understanding what specific component of the intervention was effective, a knowledge gap that is presented in other literature reviews [83]. The research designs were primarily basic experimental approaches or longitudinal studies, exposing one group of the population to the intervention versus a control group that did not receive any form of intervention. We suggest that future studies employ more robust research designs, which allow for isolating and investigating the effectiveness of different strategies (such as nudges) which may be nested within a more complex intervention. A stepped-wedge experimental approach, exposing each group gradually to a certain intervention, and measuring changes in behavior over a certain time-period [84] could be an appropriate study design to allow for this.

Secondly, a large number of studies were considered ineligible due to a lack of causality between the proposed behavior change strategy (e.g., social norms, emotions, framing) and the behavior. Hence, lacking an understanding on how the nudge strategies were eventually processed and adopted by the participants. We suggest including robust empirical verification of the proposed causality and effects of these behavior change strategies, such as pre-testing of the proposed materials and including secondary outcome measurements in the larger trial. By addressing these two main knowledge gaps, future research will be able to contribute to the evidence base concerning how nudges work and the effectiveness of different types of nudges in different contexts and among different target groups.

### Ethical standards of the included nudges

The ethical assessment of the nudges was based on the criteria developed by Engelen [36], which were adapted to fit our purpose. We did not aim to develop a novel method for assessing ethical standards more broadly within nudging for global health and health promotion, but merely included this to advance the discussion and debate concerning the ethics of nudges. Therefore, we did not label nudges as 'ethical' or 'unethical'. We simply aimed to provide a general assessment of nudges applied in the reviewed studies, by using a modified categorization from Engelen's criteria (Table 3). The findings show the majority of the included nudges were of moderate to high ethical standard (Fig 4).

Two criteria stand out, presenting nudges with a high ethical standard in the assessment: 'health benefits' and 'trust relationship'. Health benefits can be defined as measured by both the magnitude of the impact and the numbers of people positively affected by performing the behavior instigated by the nudge [36]. Since the scoping review focused on NTDs or related behaviors that prevent infectious disease transmission, hence affecting many people, it is only logical that this criterion would gain much support. However, most studies targeted behaviors rather than specific diseases, which made it difficult to quantify the health benefits. Therefore, we differentiated between these benefits based on their immediate effect and whether these depended on other factors besides the behavior of the nudgee. Nevertheless, the targeted behaviors and associated nudges remained highly beneficial for health, and therefore more legitimate to implement according to Engelen [36]. Trust relationship is the only criterion evaluating the 'nudgers', in our case the research team instigating the nudge. For this purpose, we extracted the data based on the inclusion of local collaborators and affiliation of the authors. Most studies included at least one local author, however, to our surprise, some studies did not include or acknowledge collaboration with local institutions or local authorship. Described further in the limitations, we have based this assessment on what is presented in the manuscript and did not make any efforts to contact the authors to clarify inclusion of local partners. Nevertheless, the involvement of local collaborators was assumed to be rather high, which is considered more ethical for implementing nudges.

'Reflective preferences' and 'Democratic legitimation', were two criteria which received a lower ethical assessment in the included studies. Both criteria measure the involvement by the local population in developing the study, however, focused on two different outcomes of this participation. Reflective preferences assess the involvement of the population in targeting the behavior and associated needs. Many of the included studies did not include, or only to some extent, formative research to identify and understand the context and culture of the population. Most studies decided in advance which behaviors they targeted from an epidemiological perspective. However, in order to develop more democratically legitimate nudges, formative research should also focus on the preferred behaviors from the perspective of the target population. Democratic legitimation, a criterion targeting the intervention itself, evaluates the

cultural reflection for developing the nudge interventions. The criterion indicates rather low ethical standards, which means that the nudge strategies were primarily decided upon by the research team without the involvement of local stakeholders. Developing intervention strategies using a participatory approach, is more likely to result in greater support and endorsement by the population, which may subsequently result in more ethical, legitimate and effective nudges [85].

Overall, the majority of nudges in the assessed studies had moderate to high ethical standards. This positive finding highlights awareness among researchers of ethical considerations associated with nudge strategies for behavior-based prevention and control of NTDs. We propose the criteria included in the ethical assessment as a relevant guideline for future development and research of nudge strategies in global health promotion in general. Following these proposed guidelines may foster more explicit consideration of the principles suggested by Schmidt [23], which encourages the implementation of nudges as long as these are suitably transparent and democratically controlled. Furthermore, future research could include other types of strategies, similar to nudging, such as boosting [86]. Proponents of both nudging and boosting agree that human decision making is often deficient, and that these deficits are caused by our bounded rationality. However, boosts and nudges differ mainly in 'how' they aim at improving decision making. While nudges coopt people's cognitive biases to affect behavior changes, boosts have been positioned as being more likely to empower and foster individual autonomy and competence to make their own choices, by training people to employ existing decision heuristics or new ones [87]. Boosting has been positioned by some in the field as being more in line with approaches that empower, given that they tend to be developed in a more participatory paradigm and are bottom up rather than top down. Approaches that aim to destabilize asymmetrical power relations through the co-creation of knowledge and strategies to improve health based on the lived experience of the participants themselves, are increasingly advocated [88]. Future research both with regards to boosting and nudging might benefit from greater attention to developing behavior change strategies in partnership with the target group, within a participatory approach whereby interventions are developed *with* rather than *on* populations. Such an approach would ensure interventions and strategies are firmly grounded in principles of transparency and democracy, thus reaching a higher ethical standard. Those could be used in combination with more traditional forms of nudging for global health promotion, identified by this scoping review, and foster population's health, capacities, and resilience in the long run.

## Limitations of the review

**Search strategy.**   The scoping review was designed to be as inclusive as possible, which resulted in an extensive search and several iterations of study selection. However, the nature of the literature did not allow for a simple search strategy such as others have done [27,89], where the boundary was set to studies that exclusively use the term "nudge" or "choice architecture". Interventions that include nudges have only recently been identified as promising strategies for NTD prevention and control, or more broadly infectious disease and global health promotion. By restricting our search to only those manuscripts mentioning these terms, our scoping review would lose much of its value, since few studies would meet this inclusion criterion. Moreover, there has been some tradition in global health promotion to include interventions that fall under the category nudge, although not explicitly labelling it as such (e.g., commitment, message framing). However, many of these studies use different terminologies, such as public commitment or pledge, which made it more complex for the research team to identify the interventions consistently. Nevertheless, other reviews have been reported with a broad

scope on nudge strategies, however, these have a more targeted behavioral topic, such as the self-management of chronic diseases [27]. We acknowledge that having both a broad scope on the interventions as well as the targeted behavior was highly ambitious, therefore, recognize the possibility of unidentified studies. To overcome this limitation, we published the manuscript as a pre-print, and shared it with the scientific community to allow for comments or identify possible missed studies [90].

In addition, due to the unforeseen complexity of the literature, we decided to simply focus on studies written in English. This has probably led to an underrepresentation of studies from Latin America, since relevant research has been published in Spanish. Even though most studies included an English abstract, the interventions which included nudge strategies could only be identified through full-text selection rather than title and abstract screening.

**Screening and selection.** We did not make any effort to contact authors/organizations/ actors with an incomplete description of the intervention in their study, since this would have increased the scope of the review and resulted in a significant delay of our work. We excluded all incomplete interventions and studies based on their reporting, and consequently may have missed additional relevant information. As discussed in the section on notable eligibility decisions, there is an unavoidable element of subjectivity involved, especially when considering a concept with 'fuzzy' boundaries such as nudging. Nevertheless, we believe we have taken steps to be as objective, inclusive, and transparent in our reporting as possible. We believe that the scale and nature of our screening and selection approach was appropriate given our emphasis on mapping and collating the existing evidence rather than conducting a full systematic assessment. Moreover, we have recorded and included all steps, as well the modifications to the protocol.

**Methods used for the ethical assessment.** The ethical assessment was a first attempt to establish an ethical framework for considering the inclusion of nudge strategies in behavior-based prevention and control of NTDs. This attempt fitted our intended purpose well, but it was not our aim to develop a methodology for assessing the ethics of nudge strategies across all fields. Moreover, some limitations should be taken into account when interpreting the results of this study. Firstly, the ethical assessment was based on what was described in the manuscript of the included studies. Therefore, if the authors did not mention work beyond the scope of their study, or refer to other publications, this was considered in the coding process as being missing. For example, if formative research was not mentioned, or referred to by the authors, this was considered lacking. Similarly, if local collaborators were not mentioned or acknowledged in the text, these were considered not being part of the study. We did not intend to follow up on these issues with the authors, since this was beyond the scope of our assessment. Our aim was to map what is described in the literature, not to conduct research on what could and should have been described. We do acknowledge there might have been some studies incorrectly coded. However, our aim was to present a general overview of the current ethical framework, and guidelines for meeting these ethical standards in future research.

## Conclusions

The main outcomes of this review including 67 nudges for NTD prevention and control can be summarized in three key recommendations that should inform future research when implementing nudge strategies for combatting NTDs, or global health promotion in general. Firstly, integrated health promotion approaches are currently dominating the field of behavioral NTD prevention and control efforts. Although effective for targeting a group of diseases at once, certain NTDs do not benefit from such an approach due to their specific nature and context. By altering a specific environment, nudges present an opportunity to focus on context-specific

traits of NTDs. Future efforts should include such traits and investigate the different opportunities that nudges present. Secondly, future research should aim for the application of robust study designs including rigorous process and impact evaluations which allow for a better understanding of 'what works' and 'how it works'. To date the field has primarily focused on whether or not an intervention achieves positive effects of the whole intervention package, without isolating and considering the effects of each behavior change strategy. This knowledge is necessary for achieving more targeted, and thereby more efficient and effective nudge strategies, or behavior change strategies in general. Thirdly, future research should perform rigorous ethical assessment of nudge strategies to be incorporated in an intervention, according to the aforementioned guidelines. Aim for transparency, democracy and autonomy when implementing nudges, specifically in resource constrained settings such as LMIC.

## Supporting information

**S1 Text. Search strategy in MEDLINE.**
(DOCX)

**S1 Data. Completed data extraction file including citations.**
(XLSX)

## Acknowledgments

We would like to thank research librarian Johanne Longva at the Norwegian University of Life Sciences for providing essential support in developing the search strategy and protocol for the scoping review. In addition, we wish to express our gratitude towards the two experts that provided valuable input for resolving eligibility discussions. Furthermore, we thank the members of the Department of Public Health Science (NMBU) for their constructive feedback on the review.

## Author Contributions

**Conceptualization:** Fiona Vande Velde, Hans J. Overgaard, Sheri Bastien.

**Data curation:** Fiona Vande Velde, Sheri Bastien.

**Formal analysis:** Fiona Vande Velde, Sheri Bastien.

**Funding acquisition:** Hans J. Overgaard.

**Methodology:** Fiona Vande Velde, Sheri Bastien.

**Supervision:** Sheri Bastien.

**Visualization:** Hans J. Overgaard.

**Writing – original draft:** Fiona Vande Velde, Hans J. Overgaard, Sheri Bastien.

**Writing – review & editing:** Fiona Vande Velde, Hans J. Overgaard, Sheri Bastien.

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
