## [Decision Letter · Decision Letter 0]

15 Jul 2021

Dear PhD Vande Velde,

Thank you very much for submitting your manuscript "Nudge strategies for behavior-based prevention and control of neglected tropical diseases: a scoping review and ethical assessment" for consideration at PLOS Neglected Tropical Diseases. As with all papers reviewed by the journal, your manuscript was reviewed by members of the editorial board and by several independent reviewers. In light of the reviews (below this email), we would like to invite the resubmission of a significantly-revised version that takes into account the reviewers' comments. 

We cannot make any decision about publication until we have seen the revised manuscript and your response to the reviewers' comments. Your revised manuscript is also likely to be sent to reviewers for further evaluation.

Sincerely,

Alyssa E Barry

Associate Editor

Vahid Yazdi-Feyzabadi

Deputy Editor

Reviewer's Responses to Questions

**Key Review Criteria Required for Acceptance?**

**Methods**

-Are the objectives of the study clearly articulated with a clear testable hypothesis stated?

-Is the study design appropriate to address the stated objectives?

-Is the population clearly described and appropriate for the hypothesis being tested?

-Is the sample size sufficient to ensure adequate power to address the hypothesis being tested?

-Were correct statistical analysis used to support conclusions?

-Are there concerns about ethical or regulatory requirements being met?

Reviewer #1: See attachement

Reviewer #2: - The methods are sound and compatible with the objectives to synthesis the existing body of evidence around nudging strategies within NTDs. The description of each step of the scoping review is well formulated. No statistical analysis was performed. Piloting tio ensure validity of each tool is well documented, as well the process in which the ethical framework was adjusted. 

- Do teh authors make efforts to contact the organisations/actors in the grey literature? 

- However I would make a remark here that despit

**Results**

-Does the analysis presented match the analysis plan?

-Are the results clearly and completely presented?

-Are the figures (Tables, Images) of sufficient quality for clarity?

Reviewer #1: See attachment

Reviewer #2: - There is room for improvement in terms of visualisation of the results. Figure 1 is of low quality, while figure 2 and 3 provides only basic, if not fragmented, simplification of the included nudges and then the categories based on choice architecture. 

- I appreciate the transparency in reporting the eligibility decisions taken by the review authors, including the reason of exclusion of certain study relatyed to the understanding of 'nudge' strategies as part of an intervention. There is a mistake in Reference 32 in which the sentence below referred to Vietnam while the cited study is from Indonesia. 

- There is little to be retained while reading the details on the study characteristics, especially in the 'intervention strategies', please make clear what constitues complex versus moderate used by the authors. In a similar vein, as considerable part of the included studies are actually RCT, it would be informative to include information on study design and as well (at least) outcomes of the studies included. It is somewhat moot to read about what strategies are being deployed but with little to no information on the performance/impact of such strategies. 

- For the ethical assessment, I find that there should be an improvement in reporting the outcome, is the purpose of the team to show how to apply the said framework? Currently Figire 4 gives little information in conjuction with the other sections - i would like to know which studies/which nudge strategies correspond to which ethical standard (H, M or L).

**Conclusions**

-Are the conclusions supported by the data presented?

-Are the limitations of analysis clearly described?

-Do the authors discuss how these data can be helpful to advance our understanding of the topic under study?

-Is public health relevance addressed?

Reviewer #1: See attachment

Reviewer #2: - The conclusions presented by the authors are summarized in two )- which i think quite commensurate with the data, but i think the 'robust study designs' mentioned should be elaborated further in relation with the current states found in the review.

**Editorial and Data Presentation Modifications?**

Reviewer #1: N/A

Reviewer #2: (No Response)

**Summary and General Comments**

Reviewer #1: See attachment

Reviewer #2: Thank you to the editor and the authors for the chance to review this interesting manuscript. It is well written and overall following the standard of scoping review conduct very satisfactorily.

However, I lack a bit the NTD explanation in the introduction, as well why particularly NTDs are targeted while overall most nudge strategies/studies are focusing on general health, if not child and environmental health. I have mentioned above that if the objective is to map what literature exists on the nudge strategies for NTDs, the discussions and conclusions do not go into the specificty of the findings to NTD, or in otehr words, in the end, this applies beyond NTD as well. And also on the ethical framework and assesment, I would like to see the authors comment on the aspect aligned with decolonialisation of global health, especially if the lack of autonomy, transparency etc indeed stemmed from assymetrical power relation.

PLOS authors have the option to publish the peer review history of their article (what does this mean?). If published, this will include your full peer review and any attached files.

Reviewer #1: No

Reviewer #2: No
---

## [Decision Letter · Decision Letter 1]

29 Sep 2021

Dear PhD Vande Velde,

Thank you very much for submitting your manuscript "Nudge strategies for behavior-based prevention and control of neglected tropical diseases: a scoping review and ethical assessment" for consideration at PLOS Neglected Tropical Diseases. As with all papers reviewed by the journal, your manuscript was reviewed by members of the editorial board and by several independent reviewers. The reviewers appreciated the attention to an important topic. Based on the reviews, we are likely to accept this manuscript for publication, providing that you modify the manuscript according to the review recommendations. 

Please kindly send the high quality images based on the author guideline in our journal to ensure the quality is assured.

Sincerely,

Vahid Yazdi-Feyzabadi, PhD

Deputy Editor

Vahid Yazdi-Feyzabadi

Deputy Editor

Please kindly send the high quality images based on the author guideline in our journal to ensure the quality is assured.

Reviewer's Responses to Questions

**Key Review Criteria Required for Acceptance?**

**Methods**

-Are the objectives of the study clearly articulated with a clear testable hypothesis stated?

-Is the study design appropriate to address the stated objectives?

-Is the population clearly described and appropriate for the hypothesis being tested?

-Is the sample size sufficient to ensure adequate power to address the hypothesis being tested?

-Were correct statistical analysis used to support conclusions?

-Are there concerns about ethical or regulatory requirements being met?

Reviewer #1: No further comments

**Results**

-Does the analysis presented match the analysis plan?

-Are the results clearly and completely presented?

-Are the figures (Tables, Images) of sufficient quality for clarity?

Reviewer #1: No further comments

**Conclusions**

-Are the conclusions supported by the data presented?

-Are the limitations of analysis clearly described?

-Do the authors discuss how these data can be helpful to advance our understanding of the topic under study?

-Is public health relevance addressed?

Reviewer #1: No further comments

**Editorial and Data Presentation Modifications?**

Reviewer #1: Accept

**Summary and General Comments**

Reviewer #1: The authors have done well in revising the manuscript. Nothing jumps out to me as inaccurate. Furthermore, within the journal's requirements, I think they have thoroughly covered all of the data they pulled via their scoping review. 

My only suggestion pertains to the figures and charts included in the manuscript. All the images appear to be of lower quality and are difficult to read on their own. If the authors could upload higher-quality images, this manuscript will be ready for publication and dissemination through the journal. Thank you for the opportunity to review this manuscript, and well done to the authors.

PLOS authors have the option to publish the peer review history of their article (what does this mean?). If published, this will include your full peer review and any attached files.

Reviewer #1: No

Figure Files:

Data Requirements:

Reproducibility:

References

---

## [Editor Report · Decision Letter 2]

13 Oct 2021

Dear PhD Vande Velde,

We are pleased to inform you that your manuscript 'Nudge strategies for behavior-based prevention and control of neglected tropical diseases: a scoping review and ethical assessment' has been provisionally accepted for publication in PLOS Neglected Tropical Diseases.

Best regards,

Alyssa E Barry

Associate Editor

Vahid Yazdi-Feyzabadi

Deputy Editor

---

## [Editor Report · Acceptance letter]

28 Oct 2021

Dear PhD Vande Velde,

We are delighted to inform you that your manuscript, "Nudge strategies for behavior-based prevention and control of neglected tropical diseases: a scoping review and ethical assessment," has been formally accepted for publication in PLOS Neglected Tropical Diseases.

Best regards,

Shaden Kamhawi

co-Editor-in-Chief

Paul Brindley

co-Editor-in-Chief
